# Investigating the Associations among Drug Dependents’ Family Function and Exercise Attitudes: Marital Status Differences

**DOI:** 10.3390/ijerph17218111

**Published:** 2020-11-03

**Authors:** Jianing Liu, Daniel J. McDonough, Yingying Wang, Yu Zhou, Zan Gao, Chenglin Zhou

**Affiliations:** 1School of Leisure, Shanghai University of Sport, Shanghai 200438, China; jianing0912@163.com; 2School of Kinesiology, University of Minnesota-Twin Cities, Minneapolis, MN 55455, USA; mcdo0785@umn.edu; 3School of Psychology, Shanghai University of Sport, Shanghai 200438, China; wyycris@sina.com (Y.W.); ulysses91zy@163.com (Y.Z.)

**Keywords:** behavior control, drug addiction, drug dependence, drug rehabilitation, physical activity and exercise

## Abstract

This paper examines: (1) The demographic characteristics of different marital statuses of drug dependents; (2) differences in exercise attitudes and family function by marital status; and (3) if family function factors correlated with exercise attitudes and if exercise attitude factors correlated with family function. Multivariate analyses of variance (MANOVAs) revealed significant differences in drug dependents’ exercise attitudes between married and single groups (*p* < 0.05) and the married and “other” groups (*p* < 0.01). Furthermore, we observed significant differences in drug dependents’ family function between married and single groups (*p* < 0.01) and married and other groups (*p* < 0.05). Regression analyses indicated that communication [F_change_(1,1791) = 137.819; *p* < 0.001] was a significant positive predictor for drug dependents’ exercise attitude, accounting for 7.1% of the observed variance. Moreover, 50% of the variance was explained by willingness [F_change_(1,1791) = 850.49, *p* < 0.001] and control [F_change_(1,1790) = 141.415, *p* < 0.001], which were two significant predictors of drug dependents’ family function. Findings of this study were: (1) Exercise attitude and family function of married drug dependents were better than single drug dependents and other marital status addicts; (2) communication was observed as a factor of family function that correlated with exercise attitude; and (3) willingness was related with family function.

## 1. Introduction

Drug dependence is defined as an adaptive status that develops in response to repeated drug administration and is most often revealed during withdrawal [1]. Drug dependence is a global public health concern, and accordingly, many countries and international organizations have tried to identify effective treatment strategies by which to attenuate this issue. Every year, the United Nations’ Office on Drugs and Crime updates their World Drug Report, which includes five separate components: (1) Executive summary; (2) global overview of drug demand and supply; (3) depressants; (4) stimulants; and (5) cannabis and hallucinogens [2]. Similarly, in China—a country with a particularly high prevalence of drug dependence—the China Narcotics Control Foundation releases an Annual Report on Drug Control, which focuses on drug abuse, origins of drugs, and drug trafficking [3], and disregards individuals who are dependent on drugs.

While the reasons for drug use have many inter-individual differences, a common contributor is the family dynamic. Indeed, multiple studies have found common environmental influences, particularly those within families, to influence future drug abuse behavior [4,5,6,7]. Specifically, family structure has been purported as a main underlying driver of drug abuse. For example, similar to alcoholism, Kaufman [8] observed parents’ marital status to be strongly associated with teenagers’ drug use behavior. Similarly, other scholars [9] supported these findings in that children with single parents were more likely to use and abuse drugs than those with married parents. Given marital status’ significant relation with family structure, marital status is considered as a crucial factor predicting drug use behaviors.

Although family-centered research in drug dependents has been more focused on family structure, their family *function* is also important to consider. It is well established that family function usually contains the relationships, role cognition, and problem-solving. Indeed, Lemos observed weak family relationships to be highly associated with alcohol abuse [10]. Similarly, Kaufman observed comparable findings in the role of cognition of family and the use of alcohol [8]. Moreover, coping with problems was thought to be an overt or cognitive process made available for a variety of potentially effective response strategies (e.g., orienting oneself to the problematic situation, defining the problem, generating alternative responses, making a decision, and assessing the outcome of the decision) and was added into the drug abuse prevention program in Beaulieu’s work [11]. Therefore, drug dependents’ family function needs to be explored to investigate the effects of this population’s family situation.

Physical activity is well-known for its use in the prevention and treatment of various diseases [12,13,14]. Some success has been achieved in the influence of physical activity on drug abuse prevention and treatment [15,16,17,18]. Many physical activity programs have been organized by the government or community to create opportunities for various special populations, but encouraging people to take an active part in sports has become an essential issue. Many people do not have a strong willingness to participate in exercise or lack the courage to engage in exercise behaviors. Fishbein and Ajzen proposed the Theory of Reasoned Action which examined attitudes toward behaviors [19] and has been widely used in physical activity and health promotion [20]. Manigandan was in agreeance and stated that, for most diseases, the challenge is the lack of patient compliance with exercises. Specifically, people’s perception and their attitude towards exercise matters most when determining the treatment outcome under such conditions [21]. Calvert and colleagues also attempted to use an online training tool to improve participants’ interests in exercise [22]. However, research investigating the exercise attitudes among drug dependents remains largely unexplored. Thus, exercise programs need to be explored as a drug abuse treatment.

Based on previous research regarding the families of drug dependents, family structure, and family function have become essential aspects of drug use behavior. Moreover, researchers found physical activity promotion to be an effective treatment in combatting drug dependence [23,24]. Therefore, exercise behaviors have become a priority in this line of inquiry. It is well-established that an individual’s behavior is highly correlated with their attitude. Likewise, a family’s attitude regarding behavior is attentional and develops a direct and rapid influence on each other [25]. Research has indicated that parents can have a positive or negative impact on students’ physical learning attitudes [26]. As such, the family’s attitudes towards exercise subconsciously affects individuals within the family. That said, the relationship between family function and structure and exercise attitude warrants further exploring.

Based on the review of the literature and the gaps within, the purpose of this paper was threefold: (1) To explore the demographic characteristics of different marital statuses of drug-dependents; (2) to examine whether drug dependents’ exercise attitude and family function will differ across marital status groups, with the hypothesis that exercises attitude and family function of married groups will be better than that of other marital status groups; and (3) to examine the relationship between exercise attitude and family function, with the hypothesis that role, communication, and problem-solving of family function will be highly correlated with exercise attitude, and willingness and behavioral control of exercise attitude will be highly correlated with family function.

This study has the potential to make significant contributions to this field of inquiry for several reasons. First, this study focused on family function and exercise in drug dependents, which is a population not easily accessible, and therefore, sparsely studied. Secondly, examining drug dependents’ family function is essential for their family treatment, especially exercise-based treatment. Lastly, research showing exercise to be an effective treatment/rehabilitation strategy for recovering drug dependents has compiled and these studies have mainly focused on examining the effects of exercise treatment or exploring the relevant theories. However, only a few have examined the importance of exercise attitude in this area of inquiry. Thus, our study explored the correlation among drug dependents’ exercise attitudes and other factors of family function.

## 2. Materials and Methods

### Participants

Participants were 1830 drug dependents from the Drug Rehabilitation Bureau in China (Shi Liping and Mogan Mountain in Zhejiang province, Ji Dong and Lu Zhong in Shandong province, the fifth center in Yunnan province, Baini Lake in Hunan province, Xi Shanping and Nanan in Chongqing). First, we set the inclusion criteria as follows: Non-illiteracy, no AIDS, nor head diseases. For a better investigation, we also excluded participants with symptoms of presbyopia and imitated people who were in the physiological detoxification stage of recovery. After data screening, the final sample included 1794 drug dependents (1171 males; Mage = 35.99 ± 8.54 years). Drug use mainly included methamphetamine (40.8%) and heroin (20.2%), followed by methadone (8.2%), ecstasy (8.8%), morphine (5.4%), cocaine (2.0%), adanon (10.3%), and others (4.3.%). This cross-sectional study was approved by the Ethics Committee of Shanghai University of Sport (No. 102772019RT041). Prior to study enrollment, we obtained written, informed consent from all participants.

## 3. Measures

### 3.1. Participant Demographics

To characterize the participants in this study, we first administered a self-report questionnaire which collected information on participants’ age, education level, job, salary, the age of first use drug, using time, drug use frequency, and marital status (i.e., married, single, divorced, or “other” which included same-sex cohabitation, opposite-sex cohabitation, etc.).

Descriptive statistics for the sample across each outcome and across marital status are shown in Table 1. We observed that drug dependents’ education was primarily middle school (50.6%) and middle school and below and accounted for 71.2% of the observed variance. Drug dependents’ occupations were “other” (46.5%, part-time and illegal jobs), followed by business (22.3%, e.g., small business owner), and the occupation statuses were the same way across four marital statuses. Regarding salary, 3001–5000¥ (28.1%) made up the largest percentage for the overall sample. While salary showed distinct differences across groups, compared to the total distribution (>5000 ¥, 36.5%), the other group’s salary proportion was higher (>5000 ¥, 42.6%). In the survey of drug use-related questions, participants’ first use of drugs, namely, took place between the ages of 19–25 years (42.4%), but most divorced drug dependents began to use illegal drugs at 26–35 years. Further, nearly half of the sample of drug dependents used drugs for more than five years (58.2%) and used them on a daily basis (45.2%), which was consistent among participants who were married (57.7%; 38.3%), single (57.1%; 50.1%), divorced (64.4%; 46.1%) and “other” (46.7%; 46%).

### 3.2. Outcome Measures

Exercise Attitude Questionnaire. The original version of the Exercise Attitude Questionnaire (EAQ) used a 5-point Likert-type scale to test young students’ exercise attitudes [27] A revised version of the EAQ was adapted for drug dependents (EAQ-DD), which consisted of two subscales and 10 questions in total. The EAQ has previously been used for predicting the exercise behavior of college students in China and was deemed acceptable [28]. In detail, drug dependents were asked to rate their *willingness* to engage in exercise using the following items: (a) “I’d like to spend my money on exercise”; (b) “I’ll persuade people surrounding me to exercise with me”; (c) “I always put my heart and soul into exercise”; (d) “No matter how busy I am, I can always squeeze time in to do exercise”; (e) “I’m satisfied with my persistence to exercise”; and (f) “When it is time to exercise, I cannot help but want to”. The other subscale assessed *exercise control* using the following items: (a) “I don’t know how to exercise”; (b)” It is hard for me to keep exercising when I’m tired”; (c) “I’m too lazy to exercise occasionally”; and (d) “I often fail to complete my exercise plan”. All questions were scored on a Likert scale from 1 = *totally disagree* to 5 = *totally agree*. The average scores for each subscale were used as the outcomes for exercise attitude, with higher scores indicating more positive exercise attitudes.

Family Function Scale. The FFS-DD (Drug dependents Version) was adapted from a family assessment battery (FAD), which employed a 5-point Likert scale, (1 = *strongly disagree*; 5 = *strongly agree*) and contained seven dimensions with 60 questions. Although the English version of the FAD has been widely used in both research and clinical practice, to better serve Chinese drug dependents in the Drug Rehabilitation Bureau, the FAD had to be revised [29]. The revised version (FFS-DD) has three dimensions: (1) *Role* was tested using the items: (a) “Our family often runs out of all of the things we need”; (b) “Household chores are not fully undertaken by all family members”; (c) “Our family is suffering from financial difficulties”; (d) “The family discusses who does the housework”; and (e) “Family members always need to be reminded to do something”. (2) *problem-solving* was assessed using eight items: (a) “People always know why families feel upset”; (b) “Families put things on the table”; (c) “We always speak directly to people instead of beating around the bush”; (d) “We often act on our decisions regarding problems”; (e) “After our family solves a problem, we discuss whether it has been solved”; (f) “We can solve most emotional troubles”; (g) “We face emotional problems calmly”; and (h) “We try different means to solve problems”. Lastly, *communication* was measured using the following items: (a) “We often keep our thoughts to ourselves”; and (b) “We never talk to each other when we are angry”. The FAD’s Chinese version has been used in National Conference on Sports Psychology in China (2018) and has also been used for studying the relationship between family function and adolescent problem behavior [30,31]. The average scores for each subscale were used as the outcomes for exercise attitude, with higher scores indicating more positive exercise attitudes.

### 3.3. Procedures

Data were collected from 8 Drug Rehabilitation Bureaus from February 2019 to December 2019 in 6 different provinces in China. Drug dependents completed the surveys in the gym or in the classroom with 50 people at each data collection session. Before distributing the surveys, a 5-min teaching session regarding the questionnaires and how to properly complete them was completed. Drug dependents with illiteracy completed the questionnaires with assistance from the researchers. It took approximately 15 min for drug dependents to complete the questionnaires. At last, the researchers spent 3 min checking their answers and ensuring all items were completed.

Written, informed consent was obtained from all participants before study enrollment. Following, we provided a statement of research to all participants, which detailed the intention of this study and instructed participants on how to properly answer the questions of the four different scales and demographic questionnaires. Participants’ identifications remained anonymous, and questionnaire responses were kept confidential from the police within the Drug Rehabilitation Bureaus.

### 3.4. Data Analyses

All data analyses were performed using IBM SPSS for Windows (Chicago, Il, USA, version 22). First, given both the FFS-DD and EAQ-DD were adapted versions for Chinese drug dependents, we employed an exploratory factor analysis (EFA) to confirm the validity and reliability of these scales. Following, we employed a series of multivariate analyses of variance (MANOVAs) to examine differences in drug dependents’ exercise attitudes and family function across marital status. To further elucidate differences among the marital status groups, we applied LSD post-hoc tests. Unless specified otherwise, results were expressed as mean ± standard deviation. Lastly, stepwise multiple regression analyses were employed to examine whether drug dependents’ family function factors (e.g., role, communication, problem-solving) would predict their exercise attitudes and whether willingness and control would predict exercise attitudes of drug dependents. We set the significance level to *p* < 0.05 for all analyses.

## 4. Results

### 4.1. Exploratory Factor Analysis

First, the principal component analysis of EFA as an extraction method was conducted on the pre-test data of the exercise attitude. Exercise attitude’s KMO and Bartlett’s test value was 0.853 (degrees of freedom = 45; *p* < 0.001), and in the component matrix^a^, there were six items in components 1 and 4 involved with component 2. Following rotation, the situation was maintained, and a partial correlation between variables was observed strong, and the effect of factor analysis was good for exercise attitude. Additionally, the EFA of the family function showed that the family function’s KMO and Bartlett’s test value was 0.891 (degrees of freedom = 105; *p* < 0.001) and all items were divided into three components, and when rotated, the result was the same with the component matrix^a^, the 15 variables showed a strong partial correlation and the effect of the factor analysis was good.

### 4.2. MANOVA Analysis 

The results of our MANOVAs are shown in Table 2. In detail we observed significant main effects for marital status (Wilks’ Λ = 0.989, F_6,3578_ = 3.207, *p* < 0.001, η^2^ = 0.005). Post-hoc tests revealed significant differences between the marriage and single groups (*p* < 0.05) and marriage and other groups (*p* < 0.01) of drug dependents for exercise attitude, but the marriage and divorced groups showed no statistically significant differences. Further, for drug dependents’ family function, we observed significant differences between married and single groups (*p* < 0.01) and married and other groups (*p* < 0.05), but the marriage and divorced groups showed no statistically significant differences. Moreover, the mean of the exercise attitudes in the married group was greater than that in the single group (95% confidence interval 0.03–1.40; *p* = 0.042) and other group (95% confidence interval 0.61–2.8; *p* = 0.002). The mean of the family function in the marriage group was larger than that in that in the single group (95% confidence interval 0.077–3.80; *p* = 0.003) and other group (95% confidence interval 0.07–4.88; *p* = 0.044). In addition, we observed significant differences between the distinct marital groups in willingness, role, and communication. Specifically, the willingness of the married group was significantly different than the single group (*p* < 0.01) and other group (*p* < 0.01), the role of the married group was significantly different than the single group (*p* < 0.01), and the communication of the married group was significantly greater than the other group (*p* < 0.05).

### 4.3. Regression Tests of Exercise Attitude and Family Function

The basic idea of the stepwise method is to process each regression model, step-by-step, by either adding or deleting one variable at a time based on stepping criteria [32]. The results of the stepwise regression are shown in Table 3, which indicated an exercise attitude to be highly correlated with family function. Furthermore, communication [F (1,1791) = 137.819; *p* < 0.001] was found to be a significant positive predictor of drug dependent’s exercise attitudes and accounted for 7.1% of the observed variance. Notably, role (β = −0.019; *p* = 0.456) and processing (β = 0.042; *p* = 0.15) were excluded from the model. In addition, a 50% variance was explained by willingness [F (1,1791) = 850.49, *p* < 0.001] and control [F (1,1790) = 141.415, *p* < 0.001] which were observed as two significant predictors of the family function of drug dependents.

## 5. Discussion

In this study, our two hypotheses have been confirmed using different analysis methods. The main finding was that drug dependent’s exercise attitude and family function showed significant differences across different marital statuses.

Consistent with our first hypothesis, we found significant differences among the married and other groups of drug dependents for exercise attitude and family function. Moreover, the mean score of the married group in exercise attitude and family function was higher than that of the single and other groups, which suggests that drug dependents who are married have better exercise attitudes and family function. These results are in line with Heinz’s [33] observations in that marriage has been cited as a protective factor against drug use. Indeed, Whitehead [34] also observed marital status to influence heavy drinking and found that the highest rates of heavy drinking were among divorced persons between the ages of 22 and 29 years. In our study, results showed that married and divorced groups saw no significant differences in exercise attitude or in family function. There may be two reasons for these observations: (1) Drug dependents’ families usually face stereotypes and stigma in that the family function of some married drug dependents is as bad as divorced people; and (2) the transition to being married (e.g., from single to married or from divorced to remarried) is associated with a modest reduction in fitness levels in men, while conversely, divorce is associated with modest increases, and study results suggest that these patterns may be different in women [35]. Though our study did not examine distinctions based on sex, these studies illustrate that men and women may have different trends in married and divorced groups for these outcomes.

In addition, our study found differences in drug dependents’ family communication between married and single groups. According to Stewart [36], there are significant differences between drug-abusing couples and non-drug-abusing couples in communication behaviors. As for the communication of drug dependents in married and single groups, there was no specific study available for examining this phenomenon. One study found that family function in a remarried family was highly correlated to individual family members’ illegal behavior [37]. In the present study, the married participants included both married and remarried. Therefore, it may be of concern that remarried drug dependent’s family function showed differences with the single group. Another study found that married and single drug dependents have obvious differences in family treatment effects [33]. Besides, in the drug treatment literature, researchers found that during the family treatment process, communication plays a key role [38]. Therefore, communication function within the family should not be neglected in the prophylactic and/or therapeutic phases of drug use, regardless of marital status.

Regarding drug dependents’ exercise willingness, we observed significant differences among the married, single, and other groups. Actually, few studies have focused on the relationship between marital status and exercise willingness, and the findings of these studies varied greatly. For example, Barnekow et al. observed married individuals become less active, whereas King and colleagues found the opposite [35,39,40]. From the results, there it is clear that exercise willingness is highly correlated with marital status, but there is no evidence in which an accurate conclusion on the relationship can be based. This study only found the differences in willingness in distinct marital groups, but did not discern which group’s exercise willingness was stronger. The findings provide empirical evidence for the exercise-related beliefs and attitudes in Chinese drug dependents, which helps shed new insights in this area of inquiry among this population in the developing countries [35,36].

Finally, our results indicated that, among the three dimensions of family function, communication was the only factor that could be used to predict the exercise attitude of drug dependents. Furthermore, exercise willingness was also a strong predictor of family function. Actually, there is a gap in the relationship studies between family function and exercise attitude. But with exercise becoming a popular treatment strategy for drug abuse, an increasing number of studies in the fields of sociology and/or psychology further exploring the relationships of exercise and family is needed. Based on the findings of this study, it appears possible to increase the interest in exercise or to optimize exercise treatment methods. Above all, exercise attitude holds unique value and practical implications.

In all, the results of this study provided reference values for family and exercise treatment for drug dependents. The relatively large number of participants in Mandatory Isolation Rehabilitation Center—a unique place to sample in research settings—is a major strength of this study. Our observed relationship between exercise attitude and family function shed new insights and could be used for family treatment programs and exercise treatment studies in this population. For example, in family treatment, more focus might be placed on drug dependent’s family function (i.e., communication, relationships, etc.) and exercise attitude before the employment of the kinesitherapy. Despite these strengths, there are some limitations that must be addressed. First, all of the questionnaires were collected in mandatory isolation rehabilitation centers, and therefore, could be unable to cover all kinds of drug treatment programs/strategies, and therefore may not entirely represent all Chinese drug dependents. Second, this study divided drug dependents into different marital status groups, but we did not examine sex differences, which, according to previous research, may be a major influencing factor on the explored outcomes. Lastly, marital status groups had no remarried data, and the family function score of remarried drug dependents and first-time married drug dependents may have differed.

## 6. Conclusions

In conclusion, through our investigation of a large number of drug dependents, the findings of this study dependents in mandatory isolation rehabilitation centers provide a demographic across four marital statuses. Findings suggest that exercise attitude and family function of married drug dependents is better than single drug dependents and other marital statuses. Finally, findings indicated that communication as a factor of family function can predict exercise attitude and exercise a willingness to also be a key factor in predicting the situation of family function.

## Figures and Tables

**Table 1 ijerph-17-08111-t001:** Descriptive statistics for each outcome by marital status.

Variable	Marriage	Single	Divorce	Other	Total
Age (Mean ± SD years)					
	37.76 ± 7.26	31.73 ± 7.87	40.26 ± 7.85	35.52 ± 8.61	36.01 ± 8.54
Education [*n* (%)]					
>Bachelor	11 (2)	7 (1.1)	4 (.9)	9 (6)	31(1.7)
Special college	29 (5.4)	32 (4.9)	19 (4.3)	13 (8.7)	93(5.2)
High school	113 (20.9)	142 (21.7)	103 (23)	34 (22.7)	393(21.9)
Middle School	257 (47.5)	350 (53.4)	238 (53.2)	62 (41.3)	907 (50.6)
Elementary School	121 (22.4)	118 (18)	70 (15.7)	27 (18)	336 (18.7)
illiteracy	10 (1.8)	6 (.9)	13 (2.9)	5 (3.3)	34 (1.9)
Job [*n* (%)]					
Agriculture	95 (17.6)	79 (12.1)	58 (13)	26 (17.3)	258 (14.4)
Business	171 (31.6)	90 (13.7)	93 (20.8)	45 (30)	399 (22.3)
Public Institution	23 (4.3)	27 (4.1)	35 (7.8)	5 (3.3)	90 (5)
Enterprise	41 (7.6)	68 (10.4)	46 (10.3)	14 (9.3)	169 (9.4)
Student	9 (1.7)	14 (2.1)	9 (2)	11 (7.3)	43 (2.4)
other	202 (37.3)	377 (57.6)	206 (46.1)	49 (32.7)	834 (46.5)
Salary [*n* (%)]					
None	93 (17.2)	145 (22.1)	85 (19)	24 (16)	347 (19.4)
<1000 ¥	26 (4.8)	11 (1.7)	15 (3.4)	11 (7.3)	63 (3.5)
1000–3000 ¥	57 (10.5)	79 (12.1)	64 (14.3)	24 (16)	224 (12.5)
3001–5000 ¥	139 (25.7)	207 (31.6)	131 (29.3)	27 (18)	504 (28.1)
5001–8000 ¥	104 (19.2)	111 (16.9)	83 (18.6)	38 (25.3)	336 (18.7)
>8000 ¥	122 (22.6)	102 (15.6)	69 (15.4)	26 (17.3)	319 (17.8)
First Use Age [*n* (%)]					
<12	6 (1.1)	10 (1.5)	3 (.7)	6 (4)	25 (1.4)
12–18	45 (8.3)	151 (23.1)	22 (4.9)	34 (22.7)	252 (14.1)
19–25	207 (38.3)	336 (51.3)	160 (35.8)	57 (38)	760 (42.4)
26–35	195 (36)	137 (20.9)	176 (39.4)	34 (22.7)	642 (30.2)
>35	88 (16.3)	21 (3.2)	86 (19.2)	19 (12.7)	214 (11.9)
Use time [*n* (%)]					
<2 years	33 (6.1)	38 (5.8)	26 (5.8)	18 (12)	115 (6.4)
2–3 years	105 (19.4)	116 (17.7)	62 (13.9)	31 (20.7)	314 (17.5)
4–5 years	91 (16.8)	127 (19.4)	71 (15.9)	32 (20.7)	320 (17.8)
>5 years	312 (57.7)	374 (57.1)	288 (64.4)	70 (46.7)	1044 (58.2)
Use Frequency [*n* (%)]					
Everyday	207 (38.3)	328 (50.1)	206 (46.1)	69 (46)	810 (45.2)
3–5 t/w	115 (21.3)	135 (20.6)	77 (17.2)	28 (18.7)	355 (19.8)
1–2 t/w	105 (19.4)	98 (15)	86 (19.2)	22 (14.7)	311 (17.3)
1–2 t/m	114 (21.1)	94 (14.4)	78 (17.4)	31 (20.7)	317 (17.7)

Note. Other = cohabitation of same sex, cohabitation of opposite sex, widowhood. t/w = time/week; t/m = time/month.

**Table 2 ijerph-17-08111-t002:** Group differences in exercise attitude and family function by marital status.

Variable	Total		Married		Single		Divorced		Other	
	M	SD	M	SD	M	SD	M	SD	M	SD
**Exercise Attitude**	1.1	6.04	1.64	5.77	0.93	6.26	1.06	5.95	0.06	6.09
communication	1.25	5.34	1.61	5.54	−1.05	5.24	1.38	5.29	0.47	5.09
problem-solving	3.11	5.46	3.16	5.51	2.9	5.38	3.48	5.52	2.76	5.40
role	−0.52	5.57	0.22	5.72	−1.24	5.49	−0.29	5.42	−0.71	5.45
**Family Function**	3.85	13.34	4.99	14.20	2.71	12.61	4.57	13.13	2.52	13.10
willing	1.53	5.66	2.16	5.65	1.20	5.67	1.71	5.54	0.15	5.66
control	−0.43	4.58	−0.51	4.78	−0.27	4.45	−0.65	4.43	−0.21	4.487

**Table 3 ijerph-17-08111-t003:** Regression test results of drug dependents’ exercise attitudes and family function.

Dependent Variables	Independent Variables	β	R^2^ Change	t Value
**Exercise Attitudes**				
	Role	−0.02		−0.745
	Communication	0.30	0.71	11.740 ^∗∗^
	Problem-solving	0.04		1.439
**Family Function**				
	Willing	0.49	0.322	24.913 ^∗∗^
	Control	−0.24	0.371	−11.892 ^∗∗^

Note. ∗∗ *p* < 0.001.

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
