# Peer review of "Investigating the Associations among Drug Dependents’ Family Function and Exercise Attitudes: Marital Status Differences"

_ijerph, 2020, doi:10.3390/ijerph17218111_

Round 1

Reviewer 1 Report

Major comments

  • The manuscript is well written and presented. However, authors should overall state why their study is significant and how it is different than the other published studies in the field i.e. should describe why their study is novel.

Minor Comments:

  • The manuscript needs significant improvement in grammar and syntax. It should be carefully edited to facilitate coherency.
  • Conclusion should be elaborated as well.

Author Response

Thank you for all the comments. Please see the attachment.

Reviewer 2 Report

The title is not clear: what associations are we talking about? The title have to correct.

Abstract: Drug dependence cannot be named as pandemic, because this term refers to acute infectious diseases and not to chronic diseases or mental and behavioral disorders. The reader must understand what the exercises are, because they have not been specified, so it should be indicated in the abstract whether they are physical, mathematical, etc. The authors point out that the attitude towards exercises is “usually predicted by the family factors (e.g., family structure and functioning). And most likely they are primary family factors, not the families the patient has created himself. Therefore, this connection is not clear. It would be important to find out what the attitude was in the primary family, whether the patient has taken it over and continues, or whether something has been changed in his life. The authors point out that the number of respondents is 1830, although the text reveals that the number is 1794. Then these should also be presented as a sample of the study.

Introduction part: unnecessary paragraph (lines 43 to 49), they are not relevant to this article. Old sources have been used in the introduction part, they should try and find and refer to newer ones (this century's research).

Material and methods: all respondents should be described in more detail (gender difference, average age, SD study sample (1794), not just the respondents originally interviewed. Indicate which substances have been used, for how long, and how long the remission has lasted. Part of the results part (Table 1) should be transferred to this section. Measures (3.) and Procedures (4.) should be continued as part of “Material and methods”: 2.2.; 2.3. etc.

Outcome measures: If the authors have not described how many questions were in the original survey, only stating that "used a 5-point Likert scale", the sentence on the original EAQ (119-121) does not mean anything. It is necessary to indicate which of the authors formed the EAQ-DD, a reference is required, as well as whether this survey has been adapted in a specific country and who were the authors of the adaptation, a reference is required too. The scale was adapted, not adopted (132). It is not clear whether patients were responsible for the current family or the primary family when completing the FFS-DD, or whether instructions were given for which family they should be answering the survey. The authors pointed out “4 different scales” (158), it should be clarified what has been written about (EAQ has two scales, FFS has three dimensions).

Results: As I mentioned, Table 1 is transferable to the Participants section. The table must be arranged, indicating the number of respondents for each section: marriage (n =?), Single (n =?), etc., as well as they have to indicate what it is - M, SD,% etc., this should be indicated at the top of the table. Use frequency should also be sorted, because the frequency is recurring and that does not allow readers to understand the frequency of use. It is important to indicate which substances were used by drug addicts.

Table 2 – MANOVA was used, but the data is only descriptive, authors must also specify the value of p.

Discussion: It should be indicated what these results of the study have given to clinicians, what is the contribution to treatment.

References: need to be reviewed and arranged in a uniform way. Lots of old articles, authors should find and use recent research.

Author Response

Thank you for your positive comments regarding this manuscript. This paper has been revised and each comment from the reviewer has been carefully addressed. I believe the manuscript reads better as a result. If you have any questions, please feel free to contact me.

Reviewer 3 Report

The manuscript addresses an important topic, but the authors must consider some limitations.

First of all, the abstract is far from the maximum (200 words) required by the Journal (324 words); this should be reviewed.

Regarding the introduction, it is not clear what the real reason is to investigate family relationships, with the use of drugs and sports practices. Do some studies show a cause and effect relationship between these variables? This should be better described in the text. At the end of the introduction, please insert the research hypotheses more clearly.

Regarding the method, there are several limitations.

First,  the authors did not insert inclusion criteria, only exclusion ones. Thus, it is not possible to know what kind of drugs the participants were using.

The most critical point of the work is that the instruments used were not validated in China. Consequently, the authors conducted a Factorial Exploratory Analysis (EFA) to identify the psychometric properties of the instruments. This procedure is quite mistaken because a psychometric instrument, before being validated (statistical), needs to be adapted in its population-based on rigorous procedures, according to the guidelines of the 7th Manual of the American Psychological Association. Note that this procedure has strongly biased the findings and compromised the results.

The data analyses were also not fully described. For example, EFA should not be used for reliability (Cronbach's alpha) data. Furthermore, inline 176 on page 11, the authors confused Principal Component Analysis with Exploratory Factorial Analysis. This is a misconception, and this data is outside the scope of the work, which is on a secondary plane.

In Table 1, I recommend describing the column "others," what does that mean? In Table 2, I recommend inserting the value of F and the significance level. Regarding the regression model (Table 3), it is not clear what the objective of this analysis was. I recommend re-writing this text.

In the discussion, I recommend that the authors initiate this section by recalling the main objectives of the study and the significant findings. On page 11, line 232, the authors present a secondary result that makes no sense either with the main objectives of the study or with the order of presentation of the results.

Care should also be taken with phrases that involve cause-effect relationships, as this is an exploratory study.

Author Response

(The authors gave the same response as above.)

Round 2

Reviewer 2 Report

I see the corrections have been made only partly!

The objective of the abstract Nr 3 is not understable.

The authors indicated which substances were used by 60% of respondents, but it is not clear what the other 40% used.

The frequency of use still has not been arranged in Table 1, it repeats 1-2t / w.

The authors pointed out still “4 different scales” (158), it should be clarified what has been written about (EAQ has two scales, FFS has three dimensions).

The description of the MANOVA analysis is unclear (in the Table M and SD, but in the text 95% confidence interval and p.).

References: need to be reviewed and arranged in a uniform way. Lots of old articles, you should find and use recent research.

Reviewer 3 Report

All solicitations were perormed. 

Author Response

Thank you for your positive comments regarding this manuscript. This paper has been revised and each comment from the reviewer has been carefully addressed. I believe the manuscript reads better as a result.